# Wheat Straw Incorporation Affecting Soil Carbon and Nitrogen Fractions in Chinese Paddy Soil

**Wei Dai** [1], **Jun Wang** [1], **Kaikai Fang** [1], **Luqi Cao** [2], **Zhimin Sha** [1] **and Linkui Cao** [1,*]

[1] School of Agriculture and Biology, Shanghai Jiao Tong University, 800 Dongchuan Road, Shanghai 200240, China; dw0728@sjtu.edu.cn (W.D.); junwang88@sjtu.edu.cn (J.W.); fangkaikai@sjtu.edu.cn (K.F.); zhiminsha@sjtu.edu.cn (Z.S.)

[2] Faculty of Economics and Management, East China Normal University, Shanghai 200062, China; 52174401003@stu.ecnu.edu.cn

[*] Correspondence: clk@sjtu.edu.cn

**Abstract:** Soil organic carbon (SOC) and nitrogen (N) fractions greatly affect soil health and quality. This study explored the effects of wheat straw incorporation on Chinese rice paddy fields with four treatments: (1) a control (CK), (2) a mineral NPK fertilizer (NPK), (3) the moderate wheat straw (3 t ha$^{-1}$) plus NPK (MSNPK), and (4) the high wheat straw (6 t ha$^{-1}$) plus NPK (HSNPK). In total, 0–5, 5–10, 10–20, and 20–30 cm soil depths were sampled from paddy soil in China. Compared with the CK, the HSNPK treatment ($p < 0.05$) increased the C fraction content (from 13.91 to 53.78%), mainly including SOC, microbial biomass C (MBC), water-soluble organic C (WSOC), and labile organic C (LOC) in the soil profile (0–30 cm), and it also ($p < 0.05$) increased the soil N fraction content (from 10.70 to 55.31%) such as the soil total N (TN) at 0–10 cm depth, microbial biomass N (MBN) at 0–20 cm depth, total water-soluble N (WSTN) at 0–5 and 20–30 cm depths, and total labile N (LTN) at 0–30 cm depth. The primary components of soil LOC and LTN are MBC and MBN. Various soil C and N fractions positively correlated with each other ($p < 0.05$). The HSNPK treatment promoted the soil MBC, WSOC, and LOC to SOC ratios, and also promoted MBN, WSTN, and LTN to soil TN ratios at a depth of 0–20 cm. To summarize, the application of HSNPK could maintain and improve rice paddy soil quality, which leads to increased rice grain yields.

**Keywords:** wheat straw incorporation; soil organic carbon fractions; soil nitrogen fractions; soil quality; Chinese paddy soil

## 1. Introduction

Soil organic carbon (SOC) plays a critical role in the global C cycle and dynamics [1]. Paddy fields serve as a primary source of greenhouse gases (e.g., CH$_4$ and N$_2$O) [2], and its labile C and nitrogen (N) stock changes might be linked to global greenhouse gas emissions. Fertilization application can increase rice grain yields and promote soil fertility. Improving the N fixation capacity of the soil can decrease the amount and cost of N fertilizer application and sustain the potential N supply of the soil while reducing the unfavorable impact of N loss on the environment [3]. Long-term addition of large amounts of C and N to farmland affects soil labile C and N [3]. Soil labile C and N are the most important parts of soil C and N stocks; they have far-reaching impacts on the soil C and N dynamics [4]. Soil organic matter alternations display slow degradation and are difficult to evaluate for short-term studies because of high background levels [5]. Although the content of labile organic matter is low because it is influenced by many factors (e.g., plants, microorganisms, and soil conditions), it has important meaning in soil quality changes and C/N cycling [3]. Because of the great difference in the physicochemical component and turnover times of various SOC fractions, there are great differences in the stability of soil C [6–8].

The annual crop straw yield is about 1.04 billion tons in China [9]. Crop straw burning in open fields has been the traditional method for disposal after harvest, but this leads to

loss of nutrients, organic matter, and the emission of toxic gases and greenhouse gases, consequently threatening human health and ecosystem viability [2]. Studies have proposed that crop straw is rich in organic matter, K, P, N, and microelements, and can serve as a vital natural organic fertilizer [9–11]. Incorporating crop straw into the soil improves soil fertility, promotes soil physical-structural stability, enhances soil C and N stocks, and increases crop yields [2,4,11,12]. To date, there have been many studies on the response of soil nutrients and crop yields to crop straw additions [13–16]. However, the study on the effects of wheat straw incorporation on the soil C and N fraction contents across the soil profile remains unclear, especially in rice paddy fields. The management of paddy soil is aimed at the development of an environment conducive to soil C and N formation, and strongly differs from other soil environments [17]. Because of the complexity of the paddy ecosystem, the determination of a sole organic C and N fraction cannot fully exhibit the effects of agricultural management practices (e.g., straw incorporation) on soil quality and health. Existing studies are still insufficient with regard to the changes in soil labile C and N fractions and their correlations over time with different fertilizer applications. Therefore, this study mainly focused on paddy soil in eastern China and aimed at exploring the effects of wheat straw incorporation on soil C and N accumulation at different soil depths and analyzing the relationship among different labile C and N factions. The results provide new insight for analyzing the potential mechanism of organic C and N dynamics in paddy soils under straw incorporation.

## 2. Materials and Methods

### 2.1. Study Site and Experimental Design

The field site was conducted at the Qingpu Modern Agricultural Park in Shanghai, China (31°08′ N, 121°01′ E; Figure 1). The region belongs to a subtropical monsoon climate, with a mean annual precipitation and temperature of 1056 mm and 15.5 °C, respectively. The annual daylight hours are 1960.7 h, and the frost-free days are 247 d. This region is mainly cultivated with summer rice (*Oryza sativa* L.) and winter wheat (*Triticum aestivum* L.), the basic properties of which at 0–20 cm soil depth before the experiment was begun are presented in Table 1.

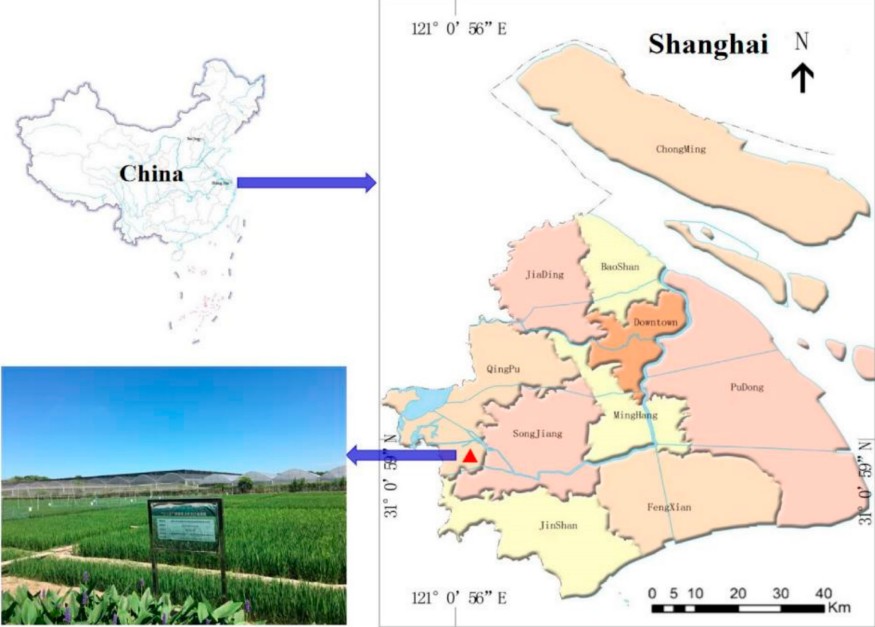

**Figure 1.** Location of the field experimental site.

**Table 1.** Basic soil properties in the 0–20 cm soil layer at the start of the experiment.

| pH | BD (g cm$^{-3}$) | SOC (g kg$^{-1}$) | TN (g kg$^{-1}$) | AN (mg kg$^{-1}$) | AP (mg kg$^{-1}$) | AK (mg kg$^{-1}$) |
|---|---|---|---|---|---|---|
| 7.08 (0.02) [†] | 1.16 (0.01) | 16.59 (0.20) | 1.96 (0.03) | 140.47 (10.71) | 36.51 (0.35) | 146.00 (3.08) |

[†] The values are expressed as means of three replicates with standard deviation in the brackets. BD, SOC, TN, AN, AP, and AK represent the bulk density, soil organic C, total N, available N, available P, and available K, respectively.

The experiment sites were established in May 2019 in a rice paddy field with four treatments: (1) CK: unfertilized control; (2) NPK: mineral NPK fertilizer applied as 300 kg N, 120 kg P, and 150 kg K ha$^{-1}$, respectively; (3) MSNPK: moderate wheat straw incorporation (3 t ha$^{-1}$) plus NPK; and (4) HSNPK: high wheat straw incorporation (6 t ha$^{-1}$) plus NPK. The treatments were set to a randomized block design in triplicate in a 56 m$^2$ plot. All wheat straw used was produced within the same fields and applied to the subsequent crop. The wheat straw was chopped to a 5–7 cm length after air-drying, and incorporated into the paddy soil at a depth of 20–30 cm using conventional tillage before the rice season. All rice straw was removed from the plots after the grain harvest. In the experimental field, rice (var. Qingjiao 307) was planted into the field on 20 June 2019 and 17 June 2020 and harvested on 3 November 2019 and 3 November 2020, respectively. The fertilizing strategy was similar to that used in local agronomic practices. Before planting the rice, basal doses of 120 kg N ha$^{-1}$, 120 kg P ha$^{-1}$, and 150 kg K ha$^{-1}$ were applied to the field. In addition, 90 kg N ha$^{-1}$ was top-dressed in the tillering and panicle initiation stages. Detailed information on agronomic management has been previously described [18].

*2.2. Soil Sampling and Analysis*

Soil samples (0–5, 5–10, 10–20, and 20–30 cm, respectively) were taken from each plot after the rice harvest on 4 November 2020. Stones and plant residues were removed, the sampled soil was mixed thoroughly, sieved through a 2 mm mesh, and divided into two portions. One portion was stored at 4 °C for the soil microbial biomass C (MBC), microbial biomass N (MBN), water-soluble organic C (WSOC), and total water-soluble N (WSTN) analysis. The other portion was air-dried for pH, SOC, and total N (TN) analysis. Soil pH was determined at a soil-to-water ratio of 1:2.5. Soil water content (SWC) was measured at the same time via soil sample collection and oven-drying at 105 °C for 48 h. The SOC and TN were obtained by dry combustion at 900 °C using an elemental analyzer (Vario EL III, CHNOS Elemental Analyzer, Elementar, Langenselbold, Germany). Inorganic C was removed by the acid neutralization method before total SOC analysis [19]. MBC and MBN were measured using the chloroform fumigation–extraction method and the C and N concentrations were measured using the Multi N/C 3100 Analyzer (Analytik Jena, Germany) [20,21]. WSOC and WSTN were measured by extracting the fresh soil samples with deionized water at a water-to-soil ratio of 5:1 and analyzing the C and N concentrations using the Multi N/C 3100 Analyzer (Analytik Jena, Germany) [22,23]. Labile organic C (LOC) was obtained by adding WSOC and MBC, and the total labile N (LTN) was obtained by adding WSTN and MBN [3]. When the rice matured on 3 November 2020, it was harvested by hand, and grain yield was measured after the grain was air-dried.

*2.3. Statistical Analysis*

Differences in soil C and N fractions among treatments were examined with a one-way ANOVA followed by Duncan's test ($p < 0.05$). Pearson's correlation analyses were identified using R software (version 3.6.2). All statistical analyses were conducted using SPSS® for Windows (version 22.0; SPSS Inc., Chicago, IL, USA), and graphs were created in Origin 2021 (Origin Lab Corporation, Northampton, MA, USA).

## 3. Results and Analysis

### 3.1. Soil pH and Water Content

Straw incorporation affected the soil pH and SWC (Figure 2). The values of soil pH in both soil depths for all treatments ranged from high to low as follows: CK > NPK > MSNPK > HSNPK (Figure 2a). This can be attributed to the acid production by decomposing organic residue of straw. The pH was ($p < 0.05$) higher in CK than in HSNPK by 8.27–8.90% at 0–10 cm depth, while no difference was found at 10–30 cm.

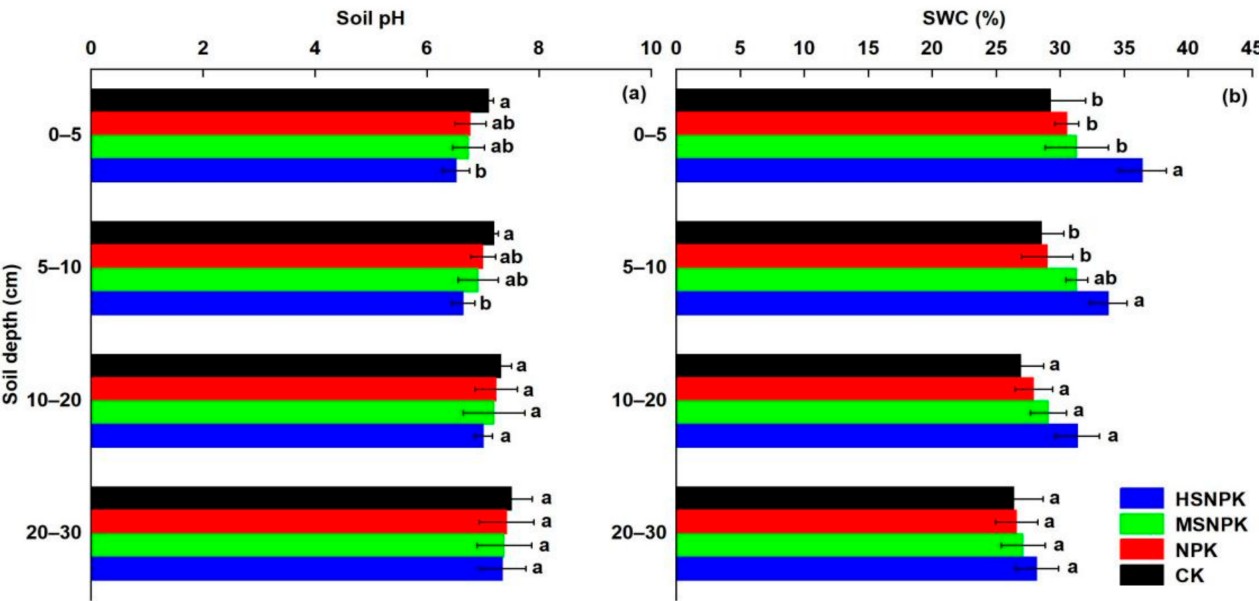

**Figure 2.** Changes in (**a**) soil pH and (**b**) SWC at different soil layers under different treatments. The values are means ± SD ($n$ = 3). Different letters mean statistically significant differences at the 0.05 level. CK: unfertilized control; NPK: mineral NPK fertilizer; MSNPK: moderate wheat straw incorporation (3 t ha$^{-1}$) plus NPK; HSNPK: high wheat straw incorporation (6 t ha$^{-1}$) plus NPK. SWC: soil water content.

Straw incorporation improved the SWC at 0–30 cm, and the SWC across different treatments exhibited the following order: HSNPK > MSNPK > NPK > CK (Figure 2b). This results in more straw input and improved SWC. The SWC was ($p < 0.05$) higher in HSNPK than in other treatments by 16.45–24.54% at 0–5 cm and in NPK by 16.56% and CK by 18.44%, respectively, at 5–10 cm, but the difference between each other was not significant at 10–30 cm depth.

### 3.2. SOC and TN

The contents of SOC and TN are related to the soil labile C and N, which are essential for analyzing soil labile C and N fraction changes. The differences in straw treatment led to significant differences in the SOC content, which ranged at both depths from high to low as follows: HSNPK > MSNPK > NPK > CK (Figure 3a). The SOC content was ($p < 0.05$) higher in HSNPK than in other treatments by 9.03–19.58% at 0–5 cm and in NPK by 8.95–11.27% at 5–20 cm and in CK by 13.91–41.46% at 5–30 cm. However, there was no significant difference between HSNPK and MSNPK at 5–30 cm. The SOC content was ($p < 0.05$) higher in MSNPK than in CK by 9.33–35.36% at 0–30 cm and in NPK by 6.79% at 10–20 cm.

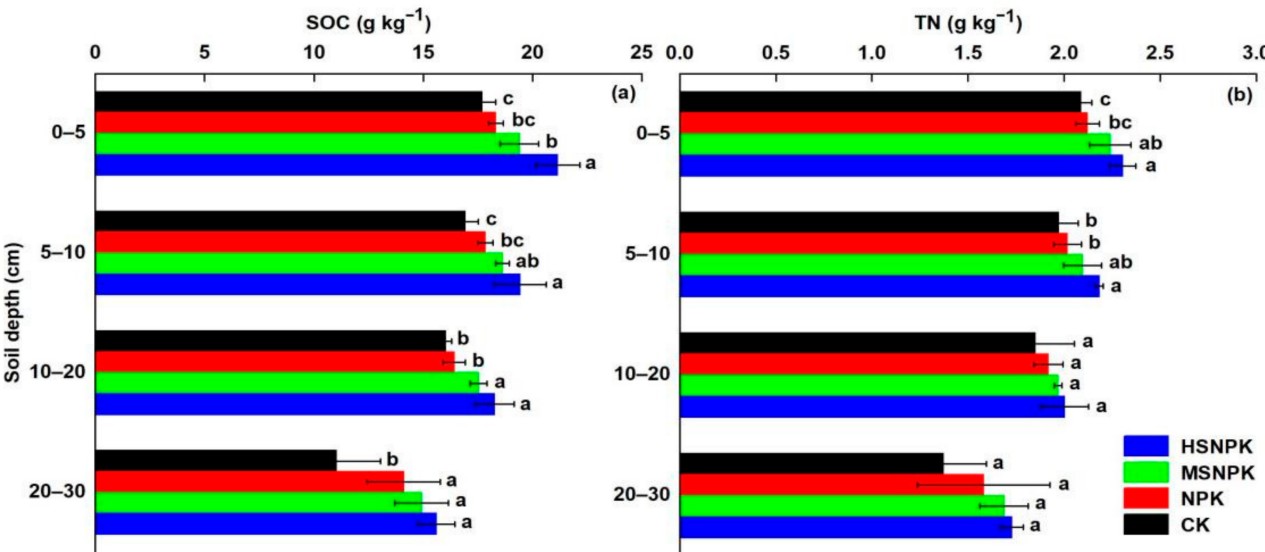

**Figure 3.** Changes in (**a**) SOC and (**b**) soil TN content at different soil layers under different treatments. The values are means ± SD (*n* = 3). Different letters mean statistically significant differences at the 0.05 level. CK: unfertilized control; NPK: mineral NPK fertilizer; MSNPK: moderate wheat straw incorporation (3 t ha$^{-1}$) plus NPK; HSNPK: high wheat straw incorporation (6 t ha$^{-1}$) plus NPK. SOC: soil organic C; TN: total N.

Soil N mainly exists in soil organic matter. Soil TN content exhibited a trend similar to that of SOC, following the order HSNPK > MSNPK > NPK > CK (Figure 3b). TN content was ($p < 0.05$) higher in HSNPK than in NPK by 8.22–8.70% and CK by 10.53–10.70%, respectively, at 0–10 cm, but did not differ among treatments at 10–30 cm. TN content was ($p < 0.05$) higher in MSNPK than in CK by 7.43% at 0–5 cm. Overall, these management practices improved soil quality and promoted the C and N sequestration potentials in rice paddy soils.

### 3.3. Soil MBC and MBN

As the key characterization indicators of soil labile C and N, MBC and MBN are more sensitive to the changes in SOC and soil TN and could also exhibit the effects of straw incorporation. The study revealed that the high rate of straw incorporation could effectively increase soil MBC content. As shown in Figure 4a, as with SOC, the contents of soil MBC in both soil depths for all treatments ranged from high to low as follows: HSNPK > MSNPK > NPK > CK. The MBC content was ($p < 0.05$) higher in HSNPK than in other treatments by 14.89–51.74% at 0–10 cm and in NPK by 26.45% at 10–20 cm and in CK by 28.71–33.52% at 10–30 cm. The MBC content was ($p < 0.05$) higher in MSNPK than in CK by 28.50% at 0–5 cm and in NPK by 18.08% and CK by 20.19%, respectively, at 10–20 cm.

The MBN content followed a pattern similar to MBC, and the increasing effect was in the order HSNPK > MSNPK > NPK > CK (Figure 4b). The MBN content was ($p < 0.05$) higher in HSNPK than in other treatments by 8.15–30.87% at 0–5 cm and in NPK by 29.13–30.38% and CK by 36.72–55.31%, respectively, at 5–20 cm, but MBN did not differ among treatments at 20–30 cm. The MBN content was ($p < 0.05$) higher in MSNPK than in NPK by 15.31% and CK by 21.00%, respectively, at 0–5 cm and in CK by 17.63% at 5–10 cm.

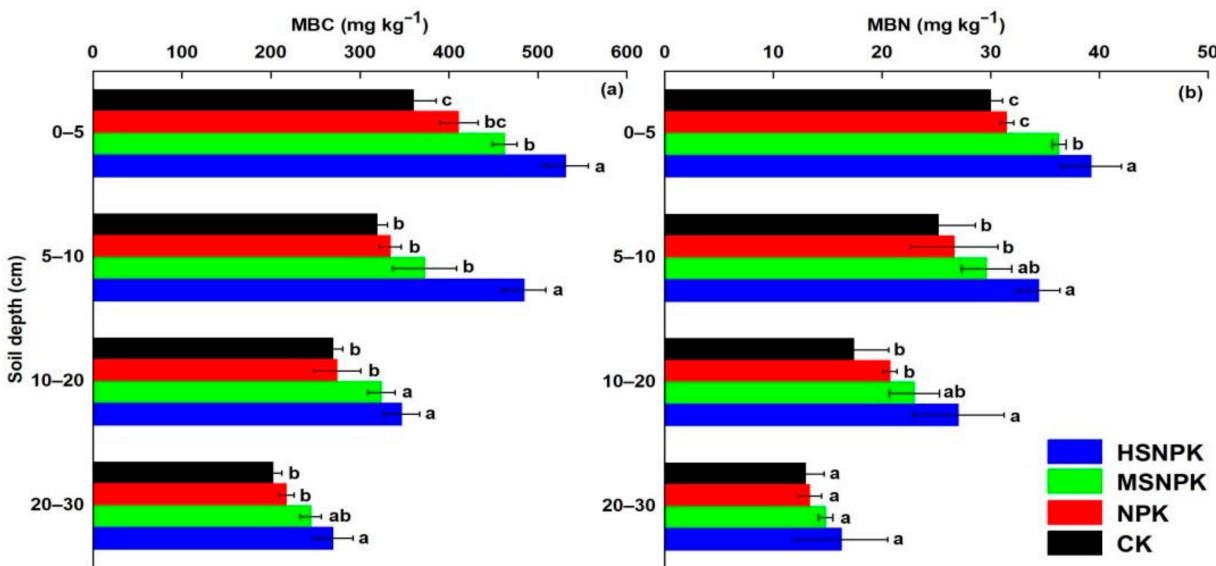

**Figure 4.** Changes in (**a**) soil MBC and (**b**) MBN content at different soil layers under different treatments. The values are means $\pm$ SD ($n$ = 3). Different letters mean statistically significant differences at the 0.05 level. CK: unfertilized control; NPK: mineral NPK fertilizer; MSNPK: moderate wheat straw incorporation (3 t ha$^{-1}$) plus NPK; HSNPK: high wheat straw incorporation (6 t ha$^{-1}$) plus NPK. MBC: microbial biomass C; MBN: microbial biomass N.

### 3.4. Soil WSOC and WSTN

The WSOC is part of the labile SOC fractions and driven by soil microorganisms [7]. WSOC content dynamics are closely related to SOC accumulation and decomposition. As the main energy source of soil microorganisms, WSOC contributes to the buffering of the desorption of soil colloids, the decomposition of litter, and the exudation of plant roots. It is highly sensitive to the loss of organic matter in the soil, so it is useful for evaluating soil quality [24]. As with SOC and MBC, the WSOC content illustrated an increasing trend of HSNPK > MSNPK > NPK > CK (Figure 5a). The WSOC content throughout the soil profile was ($p < 0.05$) higher in HSNPK than in NPK by 26.10–37.24% and CK by 35.03–53.78%, respectively. Thus, the high incorporation rate can effectively increase the WSOC content of rice paddy fields.

Similar to WSOC, the WSTN content was the highest in HSNPK, followed by MSNPK, NPK, and CK (Figure 5b). The WSTN content was ($p < 0.05$) higher in HSNPK than in other treatments by 20.56–43.35% at 0–5 cm. There was no significant difference in the WSTN content among treatments at 5–20 cm. At 20–30 cm, the WSTN content receiving the HSNPK treatment increased by 28.44% compared to the CK ($p < 0.05$).

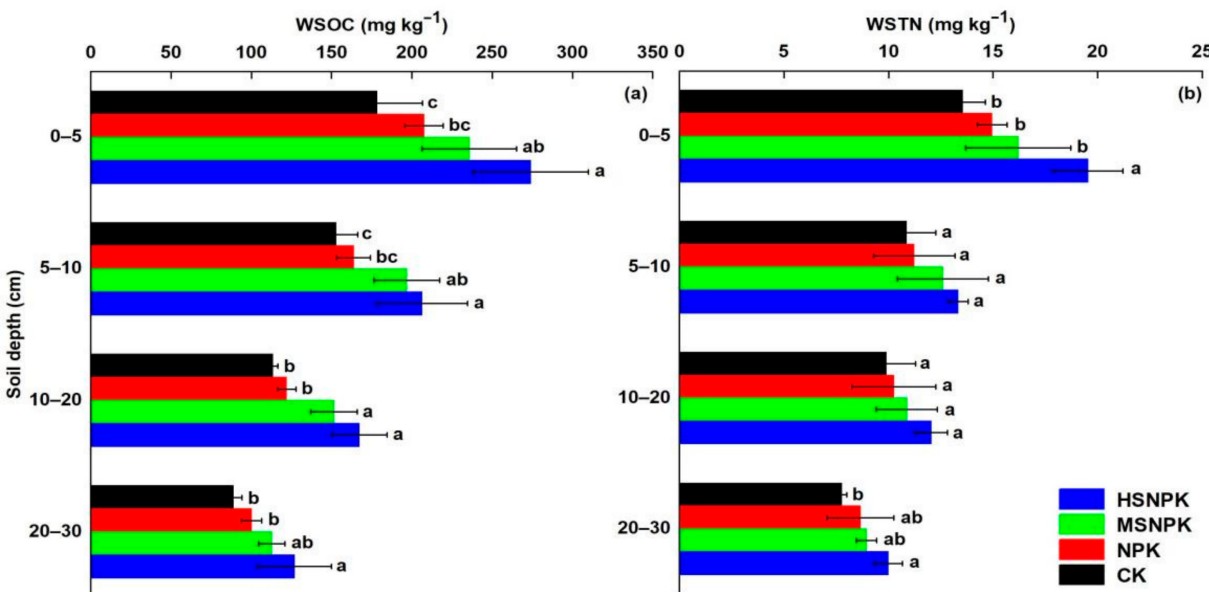

**Figure 5.** Changes in (**a**) soil WSOC and (**b**) WSTN content at different soil layers under different treatments. The values are means ± SD (*n* = 3). Different letters mean statistically significant differences at the 0.05 level. CK: unfertilized control; NPK: mineral NPK fertilizer; MSNPK: moderate wheat straw incorporation (3 t ha$^{-1}$) plus NPK; HSNPK: high wheat straw incorporation (6 t ha$^{-1}$) plus NPK. WSOC: water-soluble organic C; WSTN: total water-soluble N.

### 3.5. Soil LOC and LTN

WSOC and WSTN are better for soil microorganisms to utilize and are associated with the MBC and MBN dynamics. They are all part of labile soil C and N fractions and could be highly affected by soil ecosystems. Thus, it is not easy to utilize WSOC and WSTN as the single indicators to analyze soil LOC and LTN. By contrast, MBC and WSOC are added to stand for soil LOC and MBN and WSTN are added to stand for soil LTN in order to more accurately illustrate the effects of crop straw incorporation on the total soil labile organic C and N. The contents of MBC and MBN throughout the soil profile occupied a higher ratio of soil LOC and LTN, 70.38% and 72.11%, respectively (Figures 6 and 7). The percentages of WSOC and WSTN in the LOC and LTN were relatively smaller, at only 29.62% and 27.89%, respectively (Figures 6 and 7). Consequently, the results suggested that MBC and MBN were the most essential parts of soil labile organic C and N, and they were more highly linked to the total amount of labile organic C and N in soil.

Straw incorporation increased the soil LOC content (*p* < 0.05), and the increasing effect was in the order HSNPK > MSNPK > NPK > CK (Figure 6). The soil LOC content was (*p* < 0.05) higher in HSNPK than in other treatments by 10.98–49.67% at 0–10 and 20–30 cm (Figure 6a,b,d). At 10–20 cm, the soil LOC content was (*p* < 0.05) higher in HSNPK than in NPK by 29.77% and CK by 34.27%, respectively (Figure 6c). The soil LOC content was (*p* < 0.05) higher in MSNPK than in NPK by 12.63–20.00% and CK by 20.60–29.76%, respectively, at 0–30 cm (Figure 6).

Similar to LOC, soil LTN content was the highest in HSNPK, followed by MSNPK, NPK, and CK (Figure 7). The soil LTN content was (*p* < 0.05) higher in HSNPK than in other treatments by 11.98–35.06% at 0–5 cm (Figure 7a) and in NPK by 26.02–26.07% at 5–20 cm (Figure 7b,c) and in CK by 26.77–43.17% at 5–30 cm (Figure 7b–d). The soil LTN content was (*p* < 0.05) higher in MSNPK than in NPK by 13.07% and CK by 20.61%, respectively, 0–5 cm (Figure 7a). Overall, the high incorporation rate (*p* < 0.05) enhanced the soil labile N content.

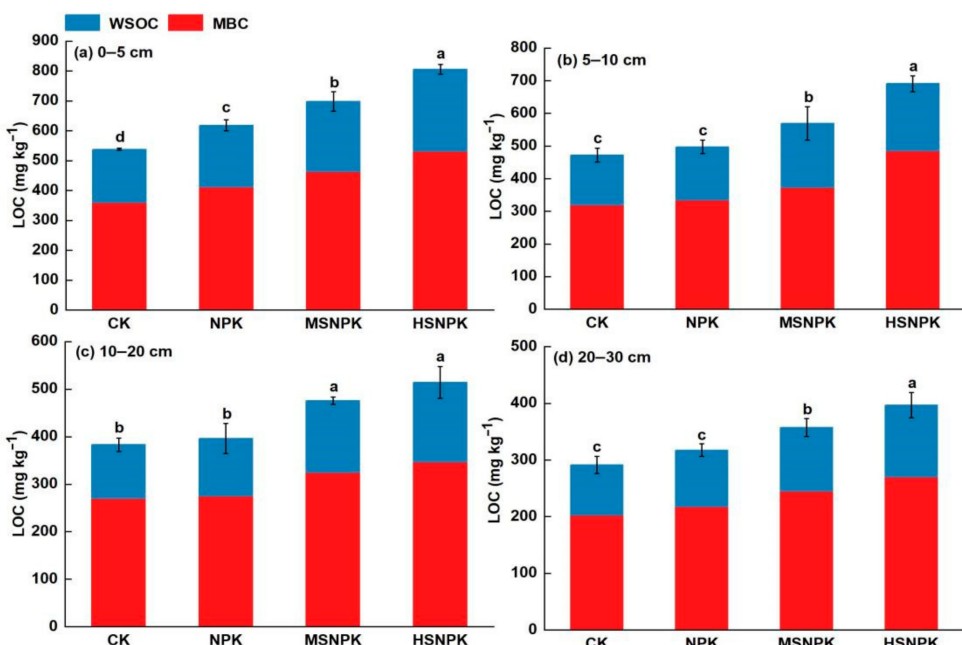

**Figure 6.** Changes in soil LOC content at different soil layers under different treatments. The values are means ± SD (*n* = 3). Different letters mean statistically significant differences at the 0.05 level. CK: unfertilized control; NPK: mineral NPK fertilizer; MSNPK: moderate wheat straw incorporation (3 t ha$^{-1}$) plus NPK; HSNPK: high wheat straw incorporation (6 t ha$^{-1}$) plus NPK. LOC: labile organic C; WSOC: water-soluble organic C; MBC: microbial biomass C.

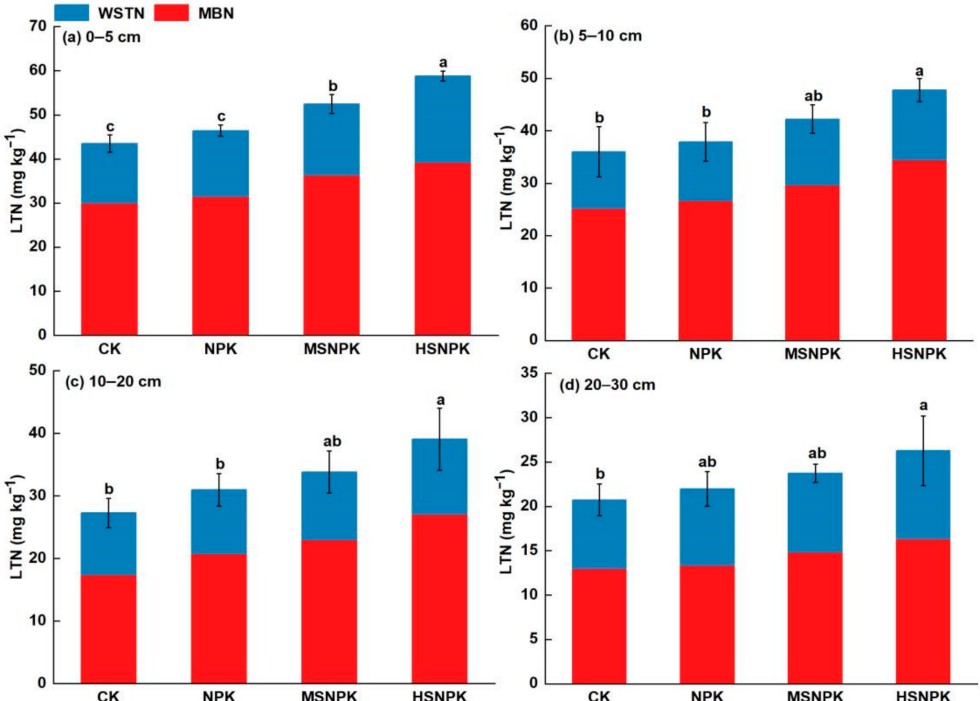

**Figure 7.** Changes in soil LTN content at different soil layers under different treatments. The values are means ± SD (*n* = 3). Different letters mean statistically significant differences at the 0.05 level. CK: unfertilized control; NPK: mineral NPK fertilizer; MSNPK: moderate wheat straw incorporation (3 t ha$^{-1}$) plus NPK; HSNPK: high wheat straw incorporation (6 t ha$^{-1}$) plus NPK. LTN: total labile N; WSTN: total water-soluble N; MBN: microbial biomass N.

### 3.6. Correlations of Soil C/N Ratio, and C and N Fractions

The proportion of the labile organic C and N fractions relative to the total SOC and TN content scan revealed the contribution of these fractions to SOC and TN. A previous study agreed that the MBC/SOC ratio could serve as a sensitive indicator of SOC change [25]. The MBC/SOC ratio is influenced by agricultural management practices (e.g., straw incorporation). Our studies indicated that the proportions of various C and N fractions to SOC and TN display differences (Table 2). The MBC/SOC ratio in HSNPK was ($p < 0.05$) higher than in CK at 0–5 cm, and there was a significant ($p < 0.05$) difference between HSNPK and the other treatments at 5–10 cm. The MBN/TN ratio was ($p < 0.05$) higher in HSNPK than in CK at 0–20 cm and NPK at 0–10 cm. The WSOC/SOC ratio in HSNPK was ($p < 0.05$) higher than in CK at 0–5 and 10–20 cm and in NPK at 10–20 cm. The WSTC/TN ratio of HSNPK was ($p < 0.05$) higher than in other treatments. The LOC/SOC ratio was ($p < 0.05$) higher in HSNPK than in NPK and CK at 0–5 and 10–20 cm, and in other treatments at 5–10 cm. The LTN/TN ratio of HSNPK was ($p < 0.05$) higher than in other treatments at 0–5 cm and in CK at 5–20 cm, and in NPK at 5–10 cm. The SOC/TN ratio was ($p < 0.05$) higher in HSNPK than in other treatments at 0–5 cm.

The SOC, TN, and different soil C and N fractions positively correlated with each other ($p < 0.01$; Figure 8). The SOC positively correlated with LOC, WSOC, and MBC ($p < 0.01$), and TN positively correlated with LTN, WSTN, and MBN ($p < 0.01$). These findings suggested that after the crop straw residues were incorporated, the increase in the accumulation of SOC mostly came from the LOC, WSOC, and MBC. Increased accumulation of soil TN was exhibited by parameters such as the LTN, WSTN, and MBN.

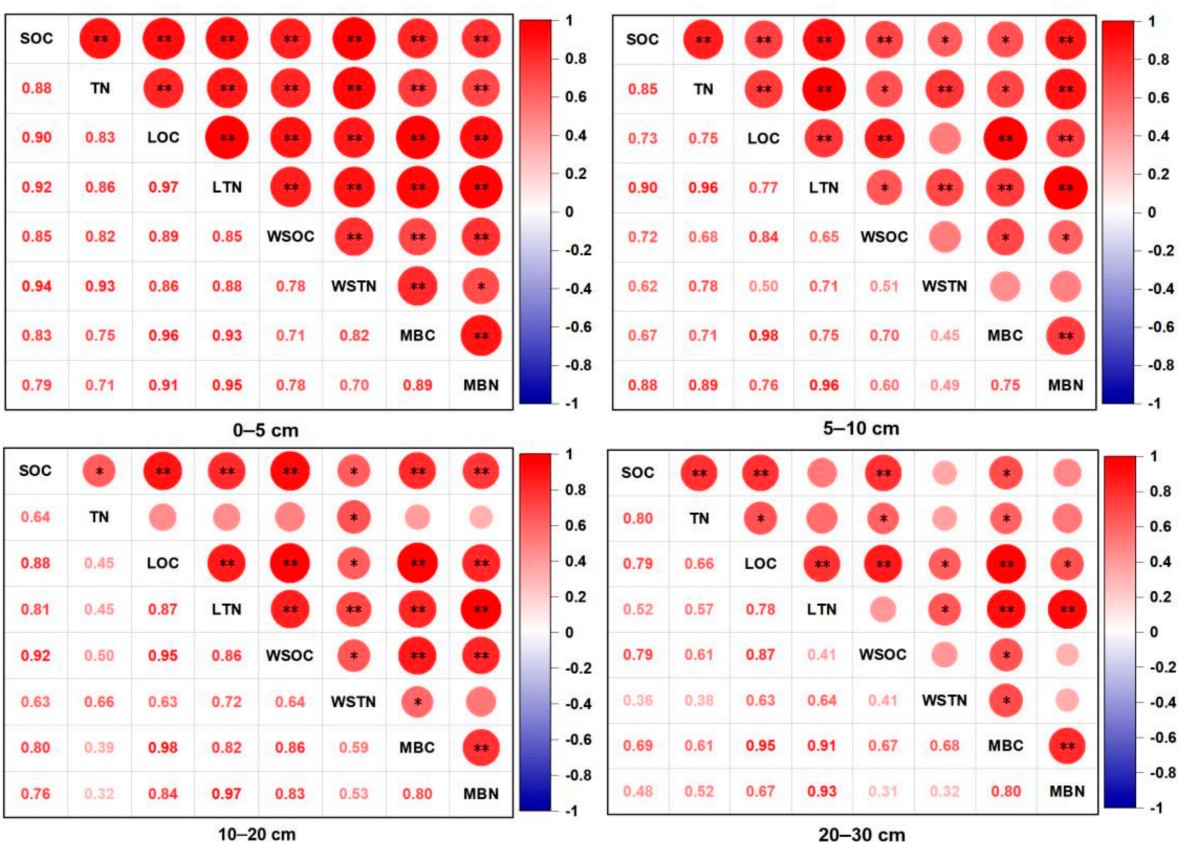

**Figure 8.** Correlation analysis of different soil C and N fractions at a depth of 0–30 cm. Red and blue circles represent positive and negative correlation, respectively. The size of the circle is proportional to the r value. *, $p < 0.05$; **, $p < 0.01$. The correlation coefficients are in the lower left panel. SOC: soil organic C; TN: total N; LOC: labile organic C; LTN: total labile N; WSOC: water-soluble organic C; WSTN: total water-soluble N; MBC: microbial biomass C; MBN: microbial biomass N.

**Table 2.** Relevant C/N ratios among different treatments at different soil layers.

| Soil Layers (cm) | Treatments | MBC/SOC (%) | MBN/TN (%) | MBC/MBN | WSOC/SOC (%) | WSTN/TN (%) | WSOC/WSTN | LOC/SOC (%) | LTN/TN (%) | SOC/TN |
|---|---|---|---|---|---|---|---|---|---|---|
| 0–5 | CK | 2.04 ± 0.16 b | 1.44 ± 0.03 b | 12.02 ± 1.10 a | 1.01 ± 0.17 b | 0.65 ± 0.03 b | 13.29 ± 2.90 a | 3.05 ± 0.10 c | 2.09 ± 0.04 c | 8.48 ± 0.08 b |
| | NPK | 2.24 ± 0.10 ab | 1.49 ± 0.06 b | 13.05 ± 0.49 a | 1.13 ± 0.06 ab | 0.70 ± 0.03 b | 13.88 ± 0.54 a | 3.38 ± 0.05 b | 2.19 ± 0.07 c | 8.64 ± 0.28 b |
| | MSNPK | 2.39 ± 0.18 a | 1.62 ± 0.09 ab | 12.74 ± 0.17 a | 1.22 ± 0.15 ab | 0.72 ± 0.08 b | 14.67 ± 1.66 a | 3.61 ± 0.26 ab | 2.34 ± 0.02 b | 8.66 ± 0.35 b |
| | HSNPK | 2.52 ± 0.21 a | 1.71 ± 0.17 a | 13.59 ± 1.27 a | 1.29 ± 0.13 a | 0.85 ± 0.05 a | 14.10 ± 2.15 a | 3.81 ± 0.10 a | 2.55 ± 0.12 a | 9.17 ± 0.24 a |
| 5–10 | CK | 1.89 ± 0.14 b | 1.27 ± 0.11 b | 12.87 ± 2.13 a | 0.90 ± 0.10 a | 0.55 ± 0.04 a | 14.30 ± 2.83 a | 2.79 ± 0.22 b | 1.82 ± 0.15 b | 8.59 ± 0.15 a |
| | NPK | 1.87 ± 0.10 b | 1.32 ± 0.18 b | 12.78 ± 2.54 a | 0.92 ± 0.06 a | 0.56 ± 0.09 a | 14.77 ± 1.74 a | 2.79 ± 0.16 b | 1.88 ± 0.13 b | 8.85 ± 0.17 a |
| | MSNPK | 2.00 ± 0.19 b | 1.41 ± 0.08 ab | 12.57 ± 0.52 a | 1.06 ± 0.09 a | 0.60 ± 0.10 a | 15.87 ± 2.59 a | 3.06 ± 0.25 b | 2.01 ± 0.04 ab | 8.90 ± 0.27 a |
| | HSNPK | 2.50 ± 0.27 a | 1.58 ± 0.07 a | 14.11 ± 1.34 a | 1.06 ± 0.15 a | 0.61 ± 0.02 a | 15.56 ± 2.73 a | 3.57 ± 0.31 a | 2.19 ± 0.08 a | 8.90 ± 0.48 a |
| 10–20 | CK | 1.68 ± 0.08 a | 0.96 ± 0.29 b | 15.75 ± 2.20 a | 0.71 ± 0.02 b | 0.53 ± 0.03 a | 11.62 ± 1.61 a | 2.39 ± 0.10 b | 1.49 ± 0.29 b | 8.73 ± 0.87 a |
| | NPK | 1.67 ± 0.14 a | 1.08 ± 0.02 ab | 13.22 ± 0.92 a | 0.74 ± 0.02 b | 0.53 ± 0.09 a | 12.15 ± 1.95 a | 2.41 ± 0.16 b | 1.61 ± 0.10 ab | 8.57 ± 0.47 a |
| | MSNPK | 1.85 ± 0.10 a | 1.17 ± 0.12 ab | 14.20 ± 1.48 a | 0.86 ± 0.07 a | 0.55 ± 0.08 a | 14.10 ± 2.09 a | 2.71 ± 0.03 a | 1.72 ± 0.18 ab | 8.90 ± 0.20 a |
| | HSNPK | 1.90 ± 0.16 a | 1.35 ± 0.17 a | 13.03 ± 1.97 a | 0.92 ± 0.07 a | 0.60 ± 0.02 a | 13.89 ± 0.75 a | 2.82 ± 0.20 a | 1.95 ± 0.19 a | 9.13 ± 0.13 a |
| 20–30 | CK | 1.87 ± 0.27 a | 0.95 ± 0.07 a | 15.73 ± 1.53 a | 0.82 ± 0.12 a | 0.58 ± 0.09 a | 11.44 ± 0.32 a | 2.69 ± 0.39 a | 1.53 ± 0.14 a | 8.03 ± 0.26 a |
| | NPK | 1.56 ± 0.21 a | 0.86 ± 0.12 a | 16.35 ± 0.80 a | 0.71 ± 0.04 a | 0.56 ± 0.14 a | 11.89 ± 2.58 a | 2.27 ± 0.25 a | 1.42 ± 0.24 a | 9.17 ± 2.12 a |
| | MSNPK | 1.65 ± 0.20 a | 0.88 ± 0.09 a | 16.54 ± 1.00 a | 0.76 ± 0.08 a | 0.53 ± 0.07 a | 12.64 ± 1.27 a | 2.41 ± 0.27 a | 1.42 ± 0.15 a | 8.84 ± 0.11 a |
| | HSNPK | 1.70 ± 0.19 a | 0.94 ± 0.24 a | 17.03 ± 2.74 a | 0.79 ± 0.10 a | 0.58 ± 0.06 a | 12.81 ± 2.96 a | 2.49 ± 0.11 a | 1.52 ± 0.23 a | 9.22 ± 0.20 a |

CK: unfertilized control; NPK: mineral NPK fertilizer; MSNPK: moderate wheat straw incorporation (3 t ha$^{-1}$) plus NPK; HSNPK: high wheat straw incorporation (6 t ha$^{-1}$) plus NPK. The values are means ± SD (*n* = 3). Different letters mean statistically significant differences at the 0.05 level. SOC: soil organic C; MBC: microbial biomass C; MBN: microbial biomass N; TN: total N; WSOC: water-soluble organic C; WSTN: total water-soluble N; LOC: labile organic C; LTN: total labile N.



### 3.7. Rice Grain Yield

Wheat straw incorporation could increase rice grain yield (Figure 9). The rice grain yields over the experimental period under CK, NPK, MSNPK, and HSNPK were 5.92, 9.20, 9.97, and 10.84 t ha$^{-1}$, respectively. Relative to the CK, the NPK, MSNPK, and HSNPK yields increased by 55.41%, 68.41%, and 83.11%, respectively, with significant differences ($p < 0.05$). The HSNPK treatment tended to further increase the rice grain yield compared with the MSNPK over the experiment period.

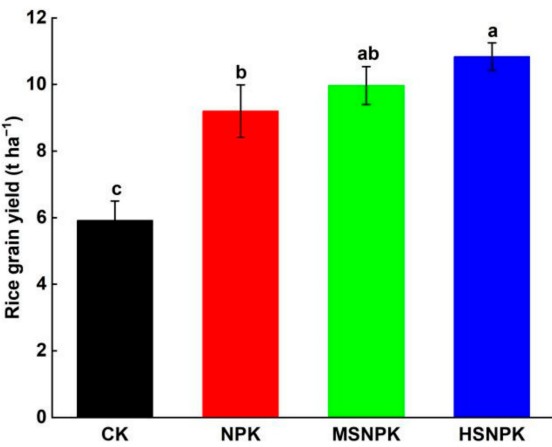

**Figure 9.** Average rice grain yield under different treatments. The values are means $\pm$ SD ($n = 3$). Different letters mean statistically significant differences at the 0.05 level. CK: unfertilized control; NPK: mineral NPK fertilizer; MSNPK: moderate wheat straw incorporation (3 t ha$^{-1}$) plus NPK; HSNPK: high wheat straw incorporation (6 t ha$^{-1}$) plus NPK.

### 4. Discussion

Crop straw incorporation could enhance soil C and N stocks and improve soil quality [3,8,26,27]. Based on the study, straw incorporation treatments increased SOC and TN content across the soil profile, especially in the high incorporation rate, indicating that more straw residue C input might have promoted the available C accumulation in the soil. The higher inputs were mainly due to the addition of organic materials (e.g., straw and roots) [1], resulting from higher crop biomass yields [28]. Similar effects were observed by Yuan et al. [7], who demonstrated that the addition of straw residues significantly increases the total SOC content.

The sole application of chemical fertilizers can promote the soil labile C stocks to some extent; however, it is as ineffective as using organic materials such as crop straw [29]. Studies [3,7,11] have revealed that straw incorporation could increase the SOC and TN contents, as well as soil labile C and N fraction contents, similar to our findings from this paddy field research. Relative to NPK and CK, straw incorporation treatments possess higher direct input of C into the soil. Another potential explanation for this behavior might be that straw incorporation promotes rice growth and increases root biomass and crop residues [3,13,28]. These are favorable for soil microbial propagation and thus boost the conversion of C and N. The studies showed that the high rate of straw incorporation treatment enhances the contents of MBC and MBN. This may be because the application of a more significant amount of straw residue could improve the physico-chemical properties of soil and thus promote the absorption and utilization of inorganic N in rice. Furthermore, it promotes the conversion of inorganic N to soil microbial biomass N, and other organic forms of N [7]. Li et al. [30] reported that the soil WSOC and MBC of straw management treatments are higher than those of the mineral fertilizer and control treatments. The study also indicated that the combined application of mineral fertilizer and crop straw incorporation produces more contents of SOC, WSOC, and MBC than the application of control or mineral fertilizer alone.



The soil MBC/SOC ratio can fully represent the ratio of labile organic C in soil [25]. This ratio reflects the difference in soil fertility from the perspective of microbiology and can be used as a good indicator to determine SOC dynamics and soil quality [3,25]. Previous studies have reported that the range of the soil MBC/SOC ratio is about 1–5%, and the input of organic materials such as crop straw could increase this ratio [30–32]. The study observed that the values of the soil MBC/SOC ratio among all treatments ranged from 1.56% to 2.52% in the entire soil profile, and the high rate of straw incorporation increased the values of the soil MBC/SOC ratio at 0–10 cm depth (Table 2). The soil MBC/MBN ratio is used to describe soil microbial community characteristics, and it can also serve as an evaluation index of soil N supply capacity and availability [33]. When the MBC/MBN ratio is small, the bioavailability of soil N is relatively high. In this study, the MBC/MBN ratio was the highest during high incorporation rate treatment at 0–10 and 20–30 cm depth, indicating that the uptake of N in rice under this treatment is larger, and thus the available N content in soil is lower. Kroer [34] found that the WSOC/WSTN ratio is inversely proportional to the bacterial growth efficiency (BGE), which reveals the proportion of C transferred from substrate (WSOM) to biomass by heterotrophic bacterial cells [4,34]. The results found that the high rate of straw incorporation decreases the WSOC/WSTN ratio and improves the soil BGE, as indicated by [4]. As for the MBN/TN and WSTN/TN ratios, the high rate treatment increased compared with the control at 0–20 cm, suggesting that this treatment is beneficial for the improvement of soil N. This might be associated with the release of N in crop straw and microbial body [3,4].

Straw incorporation contributes to rice grain yield because it increases the contents of SOC, TN, and nutrient cycling, thus promoting rice grain yield. Additionally, straw incorporation practices also enhance the soil nutrient supply, improve soil quality, decrease the adverse factors of crop growth, and strengthen the growth vigor of crops [3]. In our study, the rice grain yield was the highest in HSNPK, followed by MSNPK, NPK, and CK, and the HSNPK application tended to play a more vital role in rice grain yield increase than the application of mineral fertilizers or control alone. The findings highlighted that the application of HSNPK has the best performance in increasing rice grain yield. Similar effects were demonstrated by [14].

## 5. Conclusions

In this study, the adoption of the HSNPK treatment increased soil C and N fractions contents in the top 30 cm soil in rice paddy fields. This improvement was recorded in indicators, mainly including SOC, MBC, WSOC, and LOC, as well as the contents of soil N fractions such as soil TN, MBN, WSTN, and LTN. These indicators followed similar trends in the order HSNPK > MSNPK > NPK > CK. The primary components of soil LOC and LTN are soil MBC and MBN. The HSNPK treatment increased the MBC/SOC, WSOC/SOC, LOC/SOC, MBN/TN, WSTN/TN, and LTN/TN ratios at 0–20 cm. In summary, the application of HSNPK is a recommended management practice to sustain and improve soil quality and health in rice paddy fields, and it thereby results in promoting rice grain yield.

**Author Contributions:** Writing—original draft, review, editing, and conceptualization, W.D.; investigation, J.W.; methodology, K.F.; resources, L.C. (Luqi Cao); funding acquisition and project administration, Z.S. and L.C. (Linkui Cao). All authors have read and agreed to the published version of the manuscript.

**Funding:** This work was funded by the Shanghai Agriculture Applied Technology Development Program, China (Grant No. G20190308), and the National Key Research and Development Program of China (2016YFD0801106).

**Institutional Review Board Statement:** Not applicable.

**Informed Consent Statement:** Not applicable.

**Data Availability Statement:** Not applicable.

**Conflicts of Interest:** The authors declare no conflict of interest.

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
