# Peer review of "Wheat Straw Incorporation Affecting Soil Carbon and Nitrogen Fractions in Chinese Paddy Soil"

_agriculture, doi:10.3390/agriculture11080803_

Round 1

Reviewer 1 Report

Review to manuscript agriculture-1337726

Title: Wheat straw incorporation affecting soil carbon and nitrogen fractions in Chinese paddy soil

General comments: The manuscript describes effects of wheat straw incorporation and N fertilization on changes of carbon and nitrogen fractions in Chinese paddy soils. The topic is interesting and brings expectable results. The methodology needs substantial improvements which are indicated in Specific comments. I do not agree with the indication of SOC as it is usually a term for Soil Organic Carbon. The carbon determined at Element Analyser probably was not soil organic carbon, but total carbon. However, it depends on temperature of combustion which is not indicated. In addition, many abbreviations are not explained. In discussion I would expect larger discussion with results regarding priming effects of N and straw treatments and a possible decomposition also of original soil organic matter following NPK and straw treatments. This is a problem which is necessary to take in consideration. It was visible already in NPK treatments without straw addition, so more pronounced results were obtained under straw treatments where naturally synergic effects (straw decomposition and decomposition of original soil organic matter) could appear together. I have also no idea about the depth of straw incorporation in a soil, so there is difficult to understand if the changes in deeper soil layers are due to direct straw presence or if it can be other effect.

Specific comments:

Some abbreviations should not be well comprehensible without explanation. Add full explanation to all abbreviations when they are used for the first time – mainly SOC, MBC, WSOC, LOC, TN, MBN, WSTN, LTN... If it is not possible to give it in abstract due to maximum length given by MDPI editors, write a special subchapter Abbreviations.

Abstract

Line 14 and 16 – which parameter was taken in consideration for the percentages?  Was it total carbon?

Materials and methods

Table 1: explain which methods were used for determination of AN, AP, AK. There is difficult to understand if the given contents are really “available”.  There are many methods for determination of nutrients, some of them does not give readily available nutrient contents. Be careful in terminiology. Mainly the AN seems me to be really high. Do you mean NO3-N, NH4-N?

Line 81: The fertilization of the field seems me to be high. Were the crops able to utilize so high doses of fertilizers?  Possibly, it is normal fertilization in China, but it can cause (mainly in paddy soil) significant nitrate leaching in case that crops are not able to consume it. Were the N fertilizers divided in more doses? Also P and K doses are quite high. Which uptake of P and K by crops is determined from 1 ha? In addition, which depth and type of tillage was used?

Line 83: To which depth was incorporated the straw? There is no indication about it. The depth of incorporation is important for understanding of data obtained.

Line 88: transplanted – possibly better planted? The rice plants are transplanted from one site to other, but in the experimental field they were planted only once?

Line 89: How many years were taken in consideration. If I understand well, the experiment was established only in the year 2019. So, why … each year?

Lines 91-101: There is necessary explanation at some place the abbreviations. Naturally, they are comprehensible for skilled researchers, but there can be also young readers who can have difficulties to understand. In addition, a short characterization of methods for WSOC and WSTN should be given (water extraction?).

Line 98-99: SOC – The first I don´t believe that the abbreviation SOC is right. Authors report that they used the Element analyser. Generally, the data from these analysers are given after soil combustion under high temperatures. There exists the boundary between total C (TC) and SOC. Somebody gives it at 600 °C, somebody also at lower temperatures (450 °C). Many of analysers give data based on temperatures about 900 °C which determine also inorganic C including for instance carbonates! So, give more precise details of conditions under which the total carbon was determined (type of analyser, company, basic working conditions, namely temperature).

So, authors must to change everywhere in the text the abbreviation from SOC to TC or some other abbreviation indicating the real value of the parameter!!!

Explain also calculation of LOC and LTN! No indication is given in Materials and methods.

Results:

Lines 131-132: Possibly, the percentages for single treatments should be given for each treatment in the whole soil profile. If not, all treatments for single soil layers should be reported. However, it is difficult to write it all and the text could be probably confusing. However, the report of only selected treatment to only one soil layer is not right.

Line 133-138: The dose 300 kg N/ha was enough high to increase nitrogen content in a soil, it is necessary to take in consideration.

Lines 150-158: The same comment as in lines 131-132.

Lines 176-180: There could be also the effect of nitrate leaching and due to it higher microbial activities.

Line 187: The abbreviation LOC and LTN must be referred somewhere!

Lines 200-209: The same comment as lines 131-132 or 150-158. I suggest to simplify the text.

Discussion:

Line 271: The rice growth is not mentioned in the manuscript. I believe, that authors have data of the rice growth and yields. If authors have their own paper regarding the rice growth in the experiment, it would be nice to mention it in relation to other obtained data.

Line 283: The qMB (microbial quotient) was not mentioned in the previous text. According my meaning and general knowledge (that MBC comprises about 1-5 % of SOC) the ratio MBC/SOC (or TC) and qMB is the same. I believe that for better understanding there is better to use only the ratio used in the text. Eventually, there is necessary to explain directly in the text why the qMB is discussed.

Line 293: the highest

Line 299: The abbreviation BGE was never used before in the text. If you want to discuss it, explain the abbreviation and the necessity to discuss it in the text.

Line 312: The grain yields are not discussed in the manuscript. It is superfluous to mention them without any consequence and relation to other obtained data

Conclusions:

Line 325: the same as in the line 312.

Author Response

Response to Reviewer 1 Comments:

General comments: The manuscript describes effects of wheat straw incorporation and N fertilization on changes of carbon and nitrogen fractions in Chinese paddy soils. The topic is interesting and brings expectable results. The methodology needs substantial improvements which are indicated in Specific comments. I do not agree with the indication of SOC as it is usually a term for Soil Organic Carbon. The carbon determined at Element Analyser probably was not soil organic carbon, but total carbon. However, it depends on temperature of combustion which is not indicated. In addition, many abbreviations are not explained. In discussion I would expect larger discussion with results regarding priming effects of N and straw treatments and a possible decomposition also of original soil organic matter following NPK and straw treatments. This is a problem which is necessary to take in consideration. It was visible already in NPK treatments without straw addition, so more pronounced results were obtained under straw treatments where naturally synergic effects (straw decomposition and decomposition of original soil organic matter) could appear together. I have also no idea about the depth of straw incorporation in a soil, so there is difficult to understand if the changes in deeper soil layers are due to direct straw presence or if it can be other effect.

Our response:

We wish to thank the Reviewer’s positive comments to our manuscript, please see the revised manuscript with highlighted color.

Specific comments:

Question 1: Some abbreviations should not be well comprehensible without explanation. Add full explanation to all abbreviations when they are used for the first time–mainly SOC, MBC, WSOC, LOC, TN, MBN, WSTN, LTN... If it is not possible to give it in abstract due to maximum length given by MDPI editors, write a special subchapter Abbreviations.

Our response:

We wish to thank the reviewer’s advice, we humbly accept and agree, and we have made clear modifications in the revised manuscript. About all the abbreviations are mainly: SOC: Soil organic C; MBC: microbial biomass C; WSOC: water-soluble organic C; LOC: labile organic C; TN: total N; MBN: microbial biomass N; WSTN: total water-soluble N; LTN: total labile N; SWC: soil water content. Specifically in the revised Line 9, 15, 16, 17, 18, 19, 28, 103, 104, 105, 106, 116, and 117.

Abstract:

Question 2: Line 14 and 16–which parameter was taken in consideration for the percentages? Was it total carbon?

Our response:

As for the reviewer’s confusion, we humbly accept and carefully explain. In this study, â‘  the C fraction content (from 13.91 to 53.78%), mainly including SOC, MBC, WSOC, and LOC at 0–30 cm; â‘¡ the N fraction content (from 10.70 to 55.31%) such as the soil TN at 0–10 cm depth, MBN at 0–20 cm depth, WSTN at 0–5 and 20–30 cm depths and LTN at the 0–30 cm depth. 

Materials and methods:

Question 3: Table 1, explain which methods were used for determination of AN, AP, AK. There is difficult to understand if the given contents are really “available”. There are many methods for determination of nutrients, some of them does not give readily available nutrient contents. Be careful in terminiology. Mainly the AN seems me to be really high. Do you mean NO3--N, NH4+-N?

Our response:

According to the reviewer’s confusion, we explained it as follows:

Firstly, the measurement methods of soil available nitrogen (AN), available phosphorus (AP), available potassium (AK) are mainly: the soil AN was determined using a diffusion method and AP and AK was determined by Mo-Sb colorimetry and an NH4OAc extraction flame photometer, respectively. We also provide related measurement method reference, as follows: 

Lu, R.K. Analytical Methods for Soil and Agro-Chemistry. China Agr. Sci., Technol. 1999. (In Chinese)

In addition, because soil AN, AP, AK are the experimental ground background values, the measurement method is not mentioned in the manuscript. 

Secondly, regarding the question of available, it is mainly based on the physical and chemical expression of soil in China.

Thirdly, soil AN mainly includes inorganic N (NH4+-N, NO3--N) and easily hydrolyzed organic N. In addition, regarding the reason for the higher content (AN: 140.47 mg kg-1), we have carried out three repeated measurements on the collected soil samples in the past. The specific screenshots are as follows:

Question 4: Line 81, The fertilization of the field seems me to be high. Were the crops able to utilize so high doses of fertilizers? Possibly, it is normal fertilization in China, but it can cause (mainly in paddy soil) significant nitrate leaching in case that crops are not able to consume it. Were the N fertilizers divided in more doses? Also P and K doses are quite high. Which uptake of P and K by crops is determined from 1 ha? In addition, which depth and type of tillage was used ?

Our response:

According to the reviewer’s confusion, we humbly accept and carefully explain, and the details are as follows:

Firstly, regarding the fertilizer issue, indeed, it is normal fertilization in the Qingpu Modern Agricultural Park in Shanghai of China. Furthermore, we conducted a study on non-point source pollution in this area and found that straw incorporation into the field could effectively reduce nitrogen loss, which was not involved in this study. 

Secondly, Fertilizers were applied as 300 kg ha-1 N, 120 kg ha-1 P, and 150 kg ha-1 K, including urea (46% N), calcium superphosphate (12% P2O5), and potassium chloride (60% K2O). N fertilizer was applied at sowing stage 40%, tillering stage at 30%, and panicle stage at 30%. Both P and K fertilizers were applied before planting the rice. The detailed information on fertilization management has been previously described by Dai et al. (2021). 

Dai, W.; Gao, H.; Sha, Z.M.; Penttinen, P.; Fang, K.K.; Wang, J.; Cao, L.K. Changes in soil organic carbon fractions in response to wheat straw incorporation in a subtropical paddy field in China. J. Plant Nutr. Soil Sci. 2021, 184, 198-207.

The application standards of P fertilizer and K fertilizer are mainly calculated based on the application of Chinese rice fields.

In addition, we also revised this part, specifically in the revised Line 94-97.

Thirdly, CT, moldboard plowed to a depth of 20-30 cm soil using a SNH554 tractor (Shanghai New Holland Agriculture Machinery Co., Ltd.). Specifically in the revised Line 90-91.

Question 5: Line 83, To which depth was incorporated the straw? There is no indication about it. The depth of incorporation is important for understanding of data obtained.

Our response:

Thanks for the reviewer’s advice, and we humbly accept. Thereby, we have added straw incorporation depth information: a soil depth of 20-30 cm. Specifically in the revised Line 90.

Question 6: Line 88, transplanted–possibly better planted? The rice plants are transplanted from one site to other, but in the experimental field they were planted only once?

Our response:

Thanks for the reviewer’s advice, and we modified it. Specifically in the revised Line 92.

As for the reviewer’s confusion, we explained it as follows: Two seasons of rice have been planted from 2019 to 2020 at the experimental site. Specifically in the revised Line 92-94.

Question 7: Line 89, How many years were taken in consideration. If I understand well, the experiment was established only in the year 2019. So, why … each year?

Our response:

According to the reviewer’s confusion, we made some modifications so as to more clearly express the time of rice planting. Specifically in the revised Line 92-94.

Question 8: Lines 91-101, There is necessary explanation at some place the abbreviations. Naturally, they are comprehensible for skilled researchers, but there can be also young readers who can have difficulties to understand. In addition, a short characterization of methods for WSOC and WSTN should be given (water extraction?).

Our response:

Thanks for the reviewer’s advice, we humbly accept and agree, and we revised this part, specifically in the revised Line 103-106,113-116, and 116-117.

Question 9: Line 98-99, SOC–The first I don´t believe that the abbreviation SOC is right. Authors report that they used the Element analyser. Generally, the data from these analysers are given after soil combustion under high temperatures. There exists the boundary between total C (TC) and SOC. Somebody gives it at 600 °C, somebody also at lower temperatures (450°C). Many of analysers give data based on temperatures about 900 °C which determine also inorganic C including for instance carbonates! So, give more precise details of conditions under which the total carbon was determined (type of analyser, company, basic working conditions, namely temperature). So, authors must to change everywhere in the text the abbreviation from SOC to TC or some other abbreviation indicating the real value of the parameter!!!

Our response:

According to the reviewer’s comments, we humbly accept and agree, and we have made clear modifications in the section of soil sampling and analysis. Specifically in the revised Line 108-111.

Question 10: Explain also calculation of LOC and LTN! No indication is given in Materials and methods.

Our response:

We wish to thank the reviewer’s advice. Therefore, and we have added the related information in the revised manuscript. Specifically in the revised Line 116-117.

Results:

Question 11: Lines 131-132, Possibly, the percentages for single treatments should be given for each treatment in the whole soil profile. If not, all treatments for single soil layers should be reported. However, it is difficult to write it all and the text could be probably confusing. However, the report of only selected treatment to only one soil layer is not right.

Our response:

We thanks the reviewer’s suggestion, and we modified this part in the revised manuscript. Specifically in the revised Line 152-155.

Question 12: Line 133-138, The dose 300 kg N/ha was enough high to increase nitrogen content in a soil, it is necessary to take in consideration.  

Our response:

Thanks for the reviewer’s advice, we humbly accept. However, in this study, we used a paddy filed experiment consisting of different rates of straw incorporation under the same chemical fertilizers (NPK fertilizers) application in China, and the objectives were to explore the effects of straw incorporation on soil C and N fractions. Since the amount of fertilizer applied in every treatment was equal, and thus we did not consider the role of fertilizer in this study.

In addition, we also modified this part in the revised manuscript. Specifically in the revised Line 160. 

Question 13: Lines 150-158, The same comment as in lines 131-132. 

Our response:

We humbly accept, and we revised this part, specifically in the revised Line 177-179 and 184-186.

Question 14: Lines 176-180, There could be also the effect of nitrate leaching and due to it higher microbial activities.

Our response:

Thanks for the reviewer’s advice, and we humbly accept and understand. The role of wheat straw incorporation into the paddy field is mainly considered here, but the causes of nitrogen leaching are not taken into account. I hope that future studies will continue to explore these factors.

Question 15: Line 187, The abbreviation LOC and LTN must be referred somewhere!

Our response:

We humbly accept, and we revised the abbreviation LOC and LTN, specifically in the revised Line 16, 19, and 116-117.

Question 16: Lines 200-209, The same comment as lines 131-132 or 150-158. I suggest to simplify the text.

Our response:

Thanks for the reviewer’s advice, and we revised this part. Specifically in the revised Line 233-235 and 240-241.

Discussion:

Question 17: Line 271, The rice growth is not mentioned in the manuscript. I believe, that authors have data of the rice growth and yields. If authors have their own paper regarding the rice growth in the experiment, it would be nice to mention it in relation to other obtained data.

Our response:

Thanks for the reviewer’s advice, and we humbly accept and agree. We have added the rice grain yield in the section of Results and analysis to ensure the rigor of the manuscript. Specifically in the revised Line 291.

Question 18: Line 283, The qMB (microbial quotient) was not mentioned in the previous text. According my meaning and general knowledge (that MBC comprises about 1-5 % of SOC) the ratio MBC/SOC (or TC) and qMB is the same. I believe that for better understanding there is better to use only the ratio used in the text. Eventually, there is necessary to explain directly in the text why the qMB is discussed.

Our response:

We are very grateful to the reviewer’s advice, we humbly accept, and we directly adopt the soil MBC/SOC ratio in the revised manuscript. Regarding the importance of soil MBC/SOC ratio, we have already described in the revised manuscript, and specifically in the revised Line 331-333.

Question 19: Line 293, …the highest.

Our response:

Modified. > Line 342

Question 20: Line 299, The abbreviation BGE was never used before in the text. If you want to discuss it, explain the abbreviation and the necessity to discuss it in the text.

Our response:

We wish to thank the reviewer’s advice, and we humbly accept. Regarding the importance of the bacterial growth efficiency (BGE), we have already described in the revised manuscript, and specifically in the revised Line 344-347.

Question 21: Line 312, The grain yields are not discussed in the manuscript. It is superfluous to mention them without any consequence and relation to other obtained data.

Our response:

Thanks for the reviewer’s advice, and we have added the rice grain yield in the section of Results and analysis and the discussion of rice grain yield in the section of Discussion. Specifically in the revised Line 291 and 357-361.

Conclusions:

Question 22: Line 325, the same as in the line 312. 

Our response:

According to the reviewer’s advice, we have already added the rice grain yield in the section of Results and analysis and the discussion of rice grain yield in the section of Discussion. Specifically in the revised Line 291 and 357-361. 

Reviewer 2 Report

This article deals with the effect of wheat straw incorporation on soil carbon and nitrogen fraction in paddy soil.

There are some comments

line 35-41. The content about the decomposition of organic matter by C/N ratio seems to be added in this part.

line 45. It seems to be good to add the effect of burning in open field on PM generation. 

Figrure 1. Please increase figure resolution to see more clearly.

Results and discussion are very clearly and well documented.

Author Response

Response to Reviewer 2 Comments:

Question 1: Line 35-41, The content about the decomposition of organic matter by C/N ratio seems to be added in this part.

Our response:

We wish to thank the reviewer’s advice, but the content of labile organic matter  statements here mainly cite Yu’s paper, which emphasizes the importance of labile organic carbon. Its references are:

Yu, Q.G.; Hu, X.; Ma, J.W.; Ye, J.; Sun, W.C.; Wang, Q.; Lin, H. Effects of long-term organic material applications on soil carbon and nitrogen fractions in paddy fields. Soil Till. Res. 2020, 196, 104483.

Question 2: Line 45, It seems to be good to add the effect of burning in open field on PM generation. 

Our response:

As for the reviewer’s confusion, we humbly accept and agree, and we also modified this sentence. Specifically in the revised Line 46.

Question 3: Figure 1. Please increase figure resolution to see more clearly.

Our response:

We exported JPG format of our figures with 600 dpi from Origin 2021 (Origin Lab Corporation, Northampton, USA), which could guarantee the high quality of Figure 1. The low readable situation probably related to the conversion process for PDF format.

Results and discussion are very clearly and well documented.

Our response:

we again thank the review’s positive comment which have improved the quality of our manuscript.
